# Measuring Repositioning in Home Care for Pressure Injury Prevention and Management

**DOI:** 10.3390/s22187013

**Published:** 2022-09-16

**Authors:** Sharon Gabison, Nikola Pupic, Gary Evans, Elham Dolatabadi, Tilak Dutta

**Affiliations:** 1KITE Research Institute, Toronto Rehabilitation Institute—University Health Network, Toronto, ON M5G 2A2, Canada; 2Department of Physical Therapy, University of Toronto, Toronto, ON M5G 1V7, Canada; 3Institute of Biomedical Engineering, University of Toronto, Toronto, ON M5G 3G9, Canada; 4Vector Institute, Toronto, ON M5G 1M1, Canada

**Keywords:** pressure injuries, load cells, real-time monitoring, machine learning

## Abstract

Despite the widespread agreement on the need for the regular repositioning of at-risk individuals for pressure injury prevention and management, adherence to repositioning schedules remains poor in the clinical environment. The situation in the home environment is likely even worse. Our team has developed a non-contact system that can determine an individual’s position in bed (left-side lying, supine, or right-side lying) using data from a set of inexpensive load cells placed under the bed. This system was able to detect whether healthy participants were left-side lying, supine, or right-side lying with 94.2% accuracy in the lab environment. The objective of the present work was to deploy and test our system in the home environment for use with individuals who were sleeping in their own beds. Our system was able to detect the position of our nine participants with an F1 score of 0.982. Future work will include improving generalizability by training our classifier on more participants as well as using this system to evaluate adherence to two-hour repositioning schedules for pressure injury prevention or management. We plan to deploy this technology as part of a prompting system to alert a caregiver when a patient requires repositioning.

## 1. Introduction

Pressure injuries, also known as bedsores, are caused by damage to the skin and underlying soft tissue due to prolonged tissue deformation, disrupting cell homeostasis and leading to cell death and inflammatory oedema [1]. Pressure injuries, most commonly occurring over a bony prominence, can be a devastating secondary complication of immobility [2]. These injuries cause pain and suffering to those who live with them and increase caregiver burden [3]. In more severe cases, pressure injuries can lead to death due to septicaemia. The costs associated with pressure injuries are staggering; annual costs of hospital acquired pressure injuries in the United States are estimated to exceed USD 26.8 billion [4]. It is not surprising that the costs of preventing a pressure injury are much lower than the costs to manage them [5,6].

Pressure injury management requires a comprehensive interdisciplinary patient-centred approach that focuses on key behaviours including addressing nutrition, pressure offloading, maintaining an adequate moisture balance, ensuring the use of adequate support surfaces, and tracking pressure injuries [7]. Many but not all pressure injuries are preventable [8]. For example, patients who are unable to maintain adequate nutrition or hydration, are unable to be repositioned due to hemodynamic instability, or are at the end of life may develop pressure injuries [8]. A key best practice guideline for prevention and management of pressure injuries is to reposition at-risk patients every two hours to enable compressed tissues to return to their normal shape [9]. Individuals who are unable to reposition themselves often rely on caregivers to assist them. However, adherence to pressure offloading schedules has been found to be poor in the clinical environment and may be worse in the home care environment, where there is evidence that awareness of these injuries and the importance of repositioning is lacking [10]. There is a strong need for the development of sensor systems that can detect how often patients are being repositioned so that interventions to improve repositioning frequency can be evaluated objectively.

There have been a number of recent advances using machine learning for predicting an individual’s position and/or movements in bed [11,12,13]. These methods include load cells placed under the legs of a bed [12,14,15], pressure mats positioned over the mattress [13,16], temperature sensors [17], automated image analysis systems [11], single-motion inertial sensor units [18], and capacitive ECG signals [19]. However, there are limitations to each of these systems and the contexts in which they have been evaluated. For example, automated image analysis may be difficult to use for predicting an individual’s position in bed if there is occlusion of the individual by a blanket. Methods that rely on single-use inertial motion units require the sensors to be attached to an individual’s chest using adhesive that has the potential to cause skin damage for those with compromised skin integrity. Many of these methods have also been tested in controlled laboratory environments using predetermined positions [12,14,15] and do not incorporate the wide range of natural sleep positions adopted in bed, resulting in the limited generalizability of findings to the natural environment. Furthermore, participants in existing studies have been largely healthy volunteers and have not included the population who have or are at risk of developing a pressure injury.

Our team has developed a patient position detection system that is designed to detect whether an individual in bed is on their left side, on their back (supine), or on their right side. This system uses data from four load cells placed under the legs of the bed. Sensors collect force data and features are extracted from the data. Using advanced machine learning algorithms, we have been able to predict an individual’s position in bed as either left-side lying, supine or right-side lying with up to 94.2% accuracy in a controlled laboratory environment [20,21]. Our ultimate goal is to build on this position detection system to develop a prompting system that will continuously monitor a patient’s position in bed and alert a caregiver when the patient has remained in the same position for too long (e.g., two hours) so they can be repositioned regularly.

To date, our system has not been tested in the home environment. The primary objective of this study was to evaluate our system’s ability to predict whether participants were on their left side, supine, or on their right side while sleeping naturally in their own beds. The secondary objective of this study was to measure the inter-rater reliability of labelling the ground truth participant positions from visual inspections of time-lapse images captured of the individuals sleeping in their own bed.

## 2. Materials and Methods

This study was approved by the Research Ethics Board of University Health Network (Protocol# 17-5140).

### 2.1. Participants

Using convenience sampling, nine healthy participants (four males, five females) were recruited through advertisements placed throughout Toronto Rehabilitation Institute—University Health Network. The inclusion criterion was access to a single bed at home supported from four points below the bed. Interested participants contacted the study team via email or phone. Once participants were deemed eligible for the study and consented to participate, an overnight data collection session was scheduled. 

### 2.2. System Setup and Data Collection

Data were collected from participants while they slept overnight in their bed at home. Four single axis load cells (each comprising four strain gauges, model DLC902-30KG-HB, resolution of 1.8 g with an error of 0.2%, Hunan Detail Sensing Technology, Changsha, Hunan, China, arranged to create a full Wheatstone bridge) were placed under the legs of the bed. Signals from the load cells were amplified (gain of 500), filtered, and converted from analogue to digital values (5.0 VDC excitation) using a signal conditioner (GEN 5, AMTI, Watertown, MA, USA). NetForce software (version 3.5.2, AMTI, Watertown, MA, USA) was used on a PC Laptop (Thinkpad T520, Lenovo, Hong Kong, China, 2.5 GHz Intel Core i5 CPU, 4 GB RAM, 16 bit resolution) to collect and filter the data at 50 Hz. An infrared time-lapse camera mounted on a tripod at the foot of the bed captured an image of the participants every 30 s for ground truth labelling.

### 2.3. Data Processing

Vertical forces from each of the four load cells (LH: left sensor at the head of the bed, RH: right sensor at the head of the bed, LF: left sensor at the foot of the bed, RF: right sensor at the foot of the bed) were extracted and imported into MATLAB 2016b (Mathworks Inc., Natick, MA, USA). The distances between the sensors (length, *l*, and width, *w*) were used to compute the 12 features identified in Table 1.

The two-dimensional centre of mass (CoM) of the participant-bed system was calculated using Equations (1) and (2).
(1)$$CoM\_x=\frac{w}{2}\times \frac{LH+LF-RH-RF}{LH+LF+RH+RF}$$
(2)$$CoM\_y=\frac{l}{2}\times \frac{LH+RH-LF-RF}{LH+LF+RH+RF}$$

The CoM signal was lowpass filtered using a Chebyshev Type II filter to isolate the respiratory signal. The filter was applied using the filtfilt function that ensures a zero-phase shift to obtain the CoM_resp_x and CoM_resp_y. The times when maxima (tmax) and minima (tmin) occurred in the CoM_resp_x and CoM_resp_y signals were found by finding zero crossings for the first derivative of each signal. These times correspond with the end of each exhalation and inhalation respectively. The angle of the principal axis of the ellipsoid traced by the resultant CoM_resp signal relative to the positive x axis (positive angle measured clockwise) was calculated using Equation (3) for each tmax and subsequent tmin:(3)$$CoM\_resp\_ANG=arctan\left|\frac{CoM\_resp\_y_{\left(tmax\right)}-CoM\_resp\_y_{\left(tmin\right)}}{CoM\_resp\_x_{\left(tmax\right)}-CoM\_resp\_x_{\left(tmin\right)}}\right|$$

The changes from the components of the cardiac signal were isolated using an Equiripple FIR filter. The filtfilt function was used to bandpass filter the sum of the LH and RH signals ensuring a zero-phase shift.

Our data were processed similar to Wong et al. [20]. Each data point used for training/testing our machine learning classifiers was the average of a 45 s moving window that contained 2250 observations, where a new value was computed by shifting the window by 15 s.

### 2.4. Ground Truth Participant Position Labels

Three members of the research team independently reviewed the time-lapse images for each data point and assigned a label indicating the participant was in one of three positions: left-side-lying, supine, or right-side-lying for each 45 s. The most common label from the three reviewers was selected as the final ground truth label.

### 2.5. Data Analyses

#### 2.5.1. System Accuracy

A leave-one-participant-out cross-validation approach was used to evaluate the F1 scores and accuracy of all classifiers. Using this approach, the classifiers were trained on data from eight participants and tested on the remaining held-out participant. The procedure was repeated nine times (once for each participant). This validation approach was selected to maximize the number of training observations that were used for each classifier.

#### 2.5.2. Incremental Learning

An incremental learning approach was used to evaluate the ability of the classifier to adapt to the data from the left-out participant in a similar manner as described in Wong et al. [20]. The machine learning classifier was iteratively trained using different percentages of the left-out participant’s data (c, with c = 0%, 10%, 20%, or 30%). The left-out participants’ data were split into an incremental learning set and a test set with a ratio of 30:70 to ensure that the test set was the same across all incremental learning levels (c values) and that the performances could be compared. For example, a c = 20%, reflects 20% of the overall left-out participant’s data being added into the training set to customize the model for that specific participant. This 20% of the left-out participants’ data is found in the incremental learning set that holds 30% of the overall left-out participants’ data.

#### 2.5.3. Machine Learning Approach

Eight machine learning classifiers were used for analysis: AdaBoost (ADA), gradient boosting (GBC), light gradient boosting (LGB), logistic regression (LR), two multilayer perceptrons (MLP), support vector machine (SVM), and XGBoost (XGB). The first MLP model, MLP 1, was the model used by Wong et al. [18], and it is depicted in Figure 1. Table 2 depicts the structure of the model in more detail. The simplified MLP model, MLP 2, is depicted in Figure 2. Table 3 depicts the structure of the final simplified MLP model in more detail.

### 2.6. Statistical Analyses

The performance of machine learning classifiers was analysed using R Studio Version 1.2.1335 (RStudio, PBC, Boston, MA, USA) and inter-rater reliability was analysed using SPSS Statistics 23 (IBM SPSS Statistics, Armonk, NY, USA).

#### 2.6.1. Machine Learning Classifiers

A Shapiro–Wilk test and Levene’s test were used to assess for the normality and homogeneity of the mean accuracies and F1 scores of the different models, respectively. A Friedman’s ANOVA was used to determine if there were significant differences between the mean accuracies and the F1 scores of the machine learning classifiers at the c = 30% incremental learning level. Post hoc Wilcoxon signed-rank tests with Bonferroni adjustments (*p* < 0.017) were used to compare the top three performing models. 

For the top two performing models, the incremental learning levels corresponding to c = 0%, 10%, 20%, and 30% were compared using a Friedman’s ANOVA to determine how the level of incremental learning affected performance. Post hoc Wilcoxon signed-rank tests with Bonferroni adjustments were used to compare the performances of the models to their adjacent incremental learning values, i.e., 0% to 10%, 10% to 20%, 20% to 30%. In total, three comparisons were made between adjacent incremental learning levels, changing the *p*-value needed to reach significance to *p* < 0.017.

#### 2.6.2. Inter-Rater Reliability

In order to analyse the reliability of the ground truth data labelling of the three raters, an inter-rater reliability analysis was conducted using Fleiss multirater Kappa statistics for each position, for each participant, and for all combined participants and positions. The following criteria was used for reliability estimates: kappa < 0.20 representing poor agreement, kappa = 0.21–0.40 representing fair agreement, kappa = 0.41–0.60 representing moderate agreement, kappa = 0.61–0.80 representing substantial agreement, kappa = 0.81–1.00 representing almost perfect agreement [22].

## 3. Results

### 3.1. Participants

Participant demographics are shown in Table 4.

### 3.2. Machine Learning Models

The overall mean accuracy and F1 scores and their standard deviations across all nine participants for the classification of left-side lying, supine, or right-side lying at each incremental learning level for each model are represented in Table 5 and Table 6. Figure 3 and Figure 4 show the mean accuracy for each model at 30% incremental learning. Further comparisons of performance were performed at the c = 30% incremental learning level since these classifiers consistently outperformed classifiers at the other incremental learning levels.

#### 3.2.1. Mean Accuracy

The post hoc Wilcoxon signed-rank tests of the top three models (GBC, LGB, and XGB) with Bonferroni corrections (*p* < 0.017) demonstrated significant differences between the GBC and LGB models (V = 0, *p* < 0.01) and between the XGB and GBC models (V = 0, *p* < 0.01). No significant difference in the mean accuracies was found between LGB and XGB (V = 13, *p* = 0.933).

#### 3.2.2. F1 Scores

The Shapiro–Wilk and Levene’s tests demonstrated that the F1 scores were normally distributed and were not homogenous (F(7,64) = 3.682, *p* < 0.01). Significant differences between the F1 scores of the different models at c = 30% were found using Friedman’s ANOVA (χ^2^(7) = 54.148, *p* < 1 *×* 10^−8^). 

The post hoc Wilcoxon signed-rank test with Bonferroni corrections noted significant differences between the GBC and LGB models (V = 0, *p* < 0.01) and between the XGB and GBC models (V = 0, *p* < 0.01). No significant difference was found between the LGB and XGB models (V = 21, *p* = 0.910).

### 3.3. Incremental Learning Levels

Classifier performance for all incremental learning levels (c) for each of the top two models were compared with their adjacent c values (i.e., c = 0% to 10%, c = 10% to 20%, and c = 20% to 30%).

#### 3.3.1. Mean Accuracy

The assumption of normality and homogeneity was not met for the LGB and XGB models. The Shapiro–Wilk test was significant at c = 10% for the LGB and XGB models (W = 0.827; *p* = 0.0410, W = 0.802; *p* = 0.0216, respectively). Levene’s test was significant for the LGB and XGB models (F(3,32) = 12.575, *p* < 1 × 10^−4^; F(3,32) = 12.764, *p* < 1 × 10^−4^, respectively). 

A Friedman’s ANOVA demonstrated significant differences between LGB and XGB (χ^2^(3) = 27, *p* < 1 × 10^−5^, χ^2^(3) = 27, *p* < 1 × 10^−5^). Figure 5 shows a visual comparison between the different incremental learning levels for the LGB and XGB models.

The Wilcoxon signed-rank test with a Bonferroni adjustment (*p* < 0.017) noted significant differences for the LGB model for all comparisons of classifier performance (V = 0, *p* < 0.01 for all). For the XGB model, significant differences were noted for all comparisons of classifier performance (0% vs. 10%, 10% vs. 20%, V = 0, *p* < 0.01 and 20% vs. 30%: V = 1, *p* < 0.01).

#### 3.3.2. F1 Scores

The assumption of normality was met for the LGB and XGB models. The assumption of homogeneity was not met for the LGB and XGB models (F(3,32) = 10.079, *p* < 1 × 10^−^^4^, F(3,32) = 7.962, *p* < 1 × 10^−^^3^, respectively). Friedman’s ANOVA noted significant differences for both the LGB and XGB models (χ^2^(3) = 27, *p* < 1 × 10^−5^, χ^2^(3) = 27, *p* < 1 × 10^−^^5^ respectively). Figure 6 shows a visual comparison between the different incremental learning levels for the LGB and XGB models.

The post hoc Wilcoxon signed-rank test with Bonferroni adjustments noted significant differences with respect to F1 scores on incremental learning for all comparisons for the LGB model (0% vs. 10%, 10% vs. 20%, 20% vs. 30%, V = 0, *p* < 0.01) and for the XGB model (0% vs. 10%, 10% vs. 20%, 20% vs. 30%, V = 0, *p* < 0.01, 20% vs. 30%: V = 1, *p* < 0.01).

### 3.4. Inter-Rater Reliability

Figure 7 depicts a representative image of a participant lying in each of three different positions: left-side lying, supine, and right-side lying.

In total, 12,084 images were used for reliability estimates for labelling the ground truth data. 

The results for the inter-rater reliability analysis can be found in Table 7. Kappa values for individual positions and for each participant are provided along with the number of data points included in the analysis.

## 4. Discussion

This study is the first that we are aware of that has explored detecting an individual’s position while they slept in their own beds at home. This study was conducted on healthy participants to determine the feasibility of this methodology prior to testing with individuals at high risk of developing pressure injuries. We determined the participant’s true position in bed using time-lapse camera images while assessing the reliability of ground truth comparisons between three different raters. 

### 4.1. Machine Learning Models

In our first comparison looking at all the models, we concluded that the two best performing models were LGB and XGB at 30% incremental learning, where we achieved mean accuracies of 98.0% ± 1.42% and 98.1% ± 1.03%, respectively, and mean F1 scores of 0.981 ± 0.0116 and 0.982 ± 0.0030, respectively. This is an improvement from our previous study by Wong et al. [20] that achieved 94.2% accuracy in a controlled lab setting. We recommend using the LGB and XGB models at a 30% incremental learning level for future studies. 

### 4.2. Incremental Learning

When comparing the incremental learning levels between the top two models, LGB and XGB, significant differences between models were found in the mean accuracy and mean F1 scores, and ultimately, the best-performing models were at an incremental learning level of 30%. We noted that the greatest improvements in performance occurred between the 0% to 10% incremental learning levels (~61% to ~93% for mean accuracy and ~0.63 to 0.94~ for mean F1 score), and improvements between other levels were much smaller (<4% and 0.04). It is unclear if the differences between incremental learning levels above 10% are clinically significant. It may be desirable to select a model with slightly lower accuracy if it also requires less data to be collected for incremental learning to calibrate the system for each new end user. Future work will include further investigation of the practicality of the 30% incremental learning calibration process. 

### 4.3. Ground Truth Labelling

We used ground truth labelling for our machine learning model by having three members of the research team independently review the time-lapse images for each 45 s window and assigning one of three labels to indicate the participant’s position: left-side-lying, supine, or right-side-lying. We did not expect to the high level of agreement between raters (kappa 0.859^−1^) for ground truth labelling of the participant’s position since detecting an individual’s position in bed becomes challenging if the individual’s body is occluded by covers (e.g., sheets, blankets) [11]. However, the results of this study demonstrated that our raters were able to detect the individual’s position by visual inspection with surprising consistency, suggesting that a single rater is sufficient for providing ground truth labels in future work.

### 4.4. Future Work

Best practice guidelines recommend that individuals at risk of developing pressure injuries be placed in positions to offload high risk tissues, often assuming a range of positions between supine and side lying with the use of pillows and offloading devices (e.g., wedges) [23]. We are unsure how our machine learning models would perform on individuals at risk of developing pressure injuries while they are in their natural environment in positions based on best practice recommendations. We plan to undertake further training and testing of our machine learning models using participants with varying body masses and a wider range of sleeping positions with recommended offloading devices with individuals at risk of developing pressure injuries in future studies to improve the generalizability of our work and ensure that our high accuracy is not the result of overfitting.

Our ultimate goal of this work is to use our non-contact sensor system as an objective and unobtrusive tool for tracking repositioning in the home environment to gain a better understanding of how often patients are repositioned and to evaluate interventions that may improve adherence to regular repositioning. These interventions may include combinations of education, prompting systems, and pain management as well as paid caregiver support in the home. Our work may include extending the functionality of our sensor system to create a repositioning prompting system, designed to provide prompts to a caregiver when the patient has remained in the same position for too long.

## 5. Conclusions

The results of this study found that our system was able to detect the position of nine participants at home in their own beds (as either left-side lying, supine, or right-side lying,) with 98.1% accuracy and an F1 score of 0.982 using the XGBoost classifier. An inter-rater reliability analysis using Fleiss’s kappa found almost perfect agreement (κ = 0.935) between the ground truth position labels created by our three independent reviewers.

These results demonstrate tha tour system has excellent potential for use in the home environment for tracking patient position continuously for pressure injury prevention and management.

## Figures and Tables

**Figure 1 sensors-22-07013-f001:**
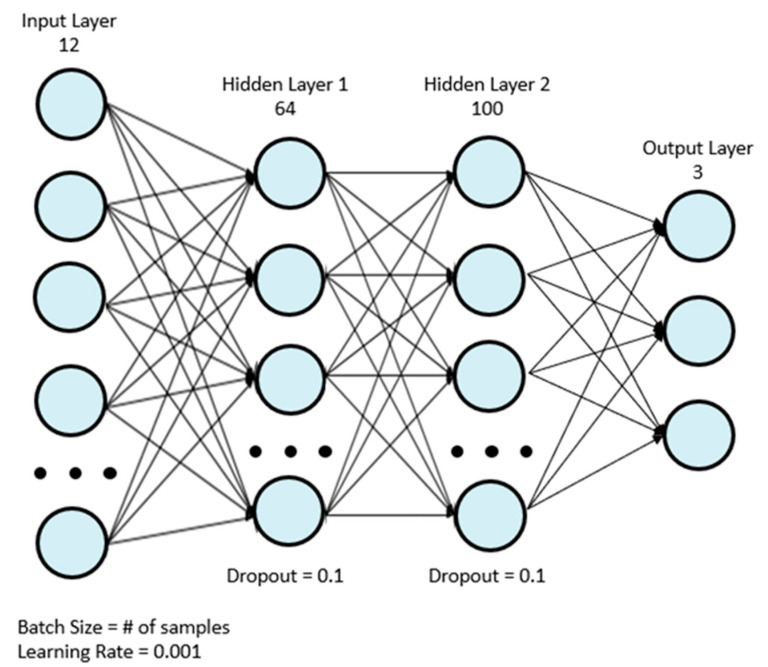
Original MLP model used in the in-lab study by Wong et al. [20].

**Figure 2 sensors-22-07013-f002:**
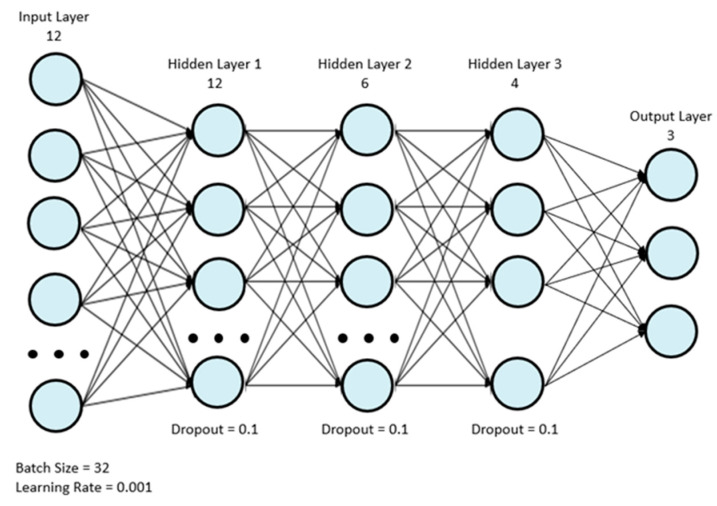
Final simplified MLP model.

**Figure 3 sensors-22-07013-f003:**
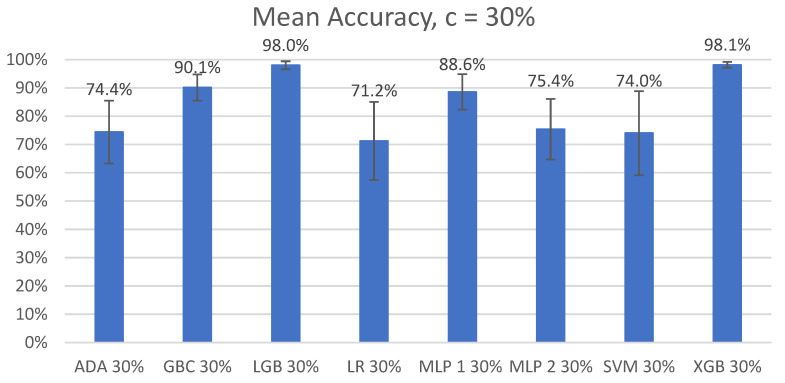
Mean accuracy of left-side lying, supine, or right-side lying classification by model for the incremental learning level c = 30%. Error bars represent standard deviations.

**Figure 4 sensors-22-07013-f004:**
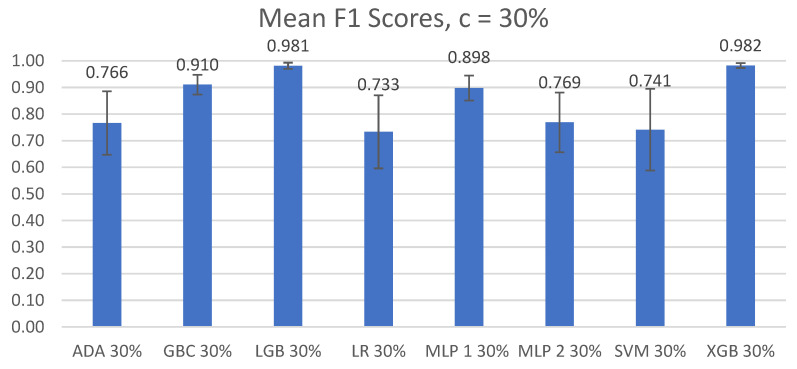
Mean F1 scores of left-side lying, supine, or right-side lying classification by model for the incremental learning level c = 30%. Error bars represent standard deviations.

**Figure 5 sensors-22-07013-f005:**
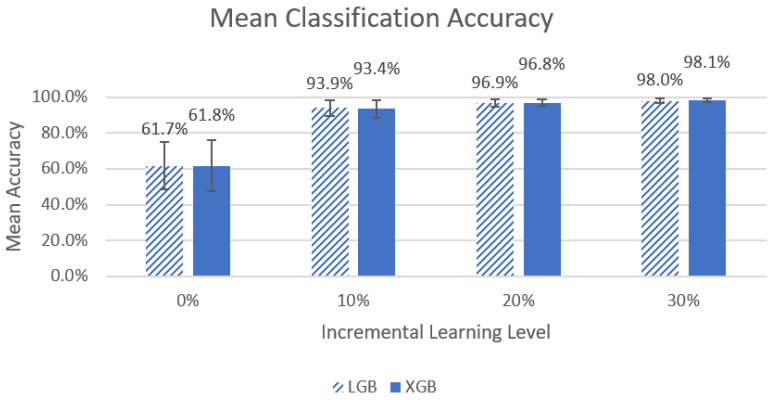
Graph of left-side lying, right-side lying, or supine classification mean accuracy for the LGB and XGB models across the four different incremental learning levels. The error bars represent standard deviation.

**Figure 6 sensors-22-07013-f006:**
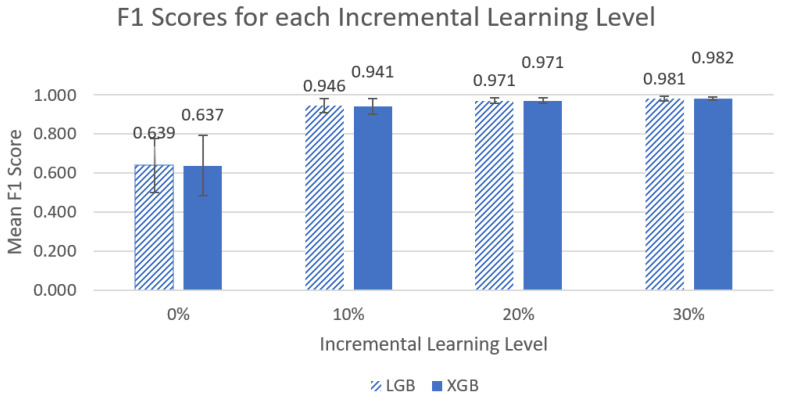
Graph of left-side lying, supine, or right-side lying classification mean F1 metrics for the LGB and XGB models across the four different incremental learning levels. The error bars represent standard deviation.

**Figure 7 sensors-22-07013-f007:**
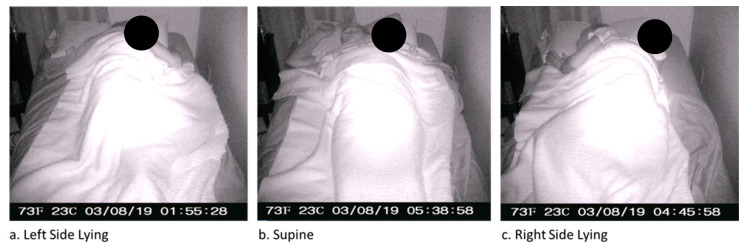
Sample time-lapse images of Participant 1.

**Table 1 sensors-22-07013-t001:** Features calculated from the load cell data.

Feature	Description
mean CoM_x	The mean of CoM_x parallel to the width of the bed
meanCoM_y	The mean of CoM_y parallel to the length of the bed
ratio_meanCoM	The quotient of meanCoM_y divided by CoM_x
stdCoM_x	The standard deviation of CoM_x
stdCoM_y	The standard deviation of CoM_y
ratio_stdCoM	The quotient of stdCoM_y divided by stdCoM_x
CoM_resp_ANG	CoM angle during inhalation phase only, averaged for all occurrences
stdCoM_resp_ANG	Standard deviation of CoM_resp_ANG
rmsCoM_resp_x	The root mean square of the x-component of CoM_resp during both inhale and exhale phases, normalized to the 97th percentile
rmsCoM_resp_y	The root mean square of the y-component of CoM_resp during both inhale and exhale phases, normalized to the 97th percentile
ratio_rmsCoM_resp	The quotient of rmsCoM_resp_y divided by rmsCoM_resp_x
rmsPulse	The root mean square of the load cell signals filtered to capture changes from the cardiac cycle

**Table 2 sensors-22-07013-t002:** Architecture of the original MLP model by Wong et al. [20].

Layer	Number of Nodes	Activation Function
Input	12	ReLu
Fully Connected 1	64	ReLu
Dropout	N/A	N/A
Batch Normalization	N/A	N/A
Fully Connected 2	100	ReLu
Dropout	N/A	N/A
Batch Normalization	N/A	N/A
Output	3	Softmax

**Table 3 sensors-22-07013-t003:** Architecture of the final simplified MLP model.

Layer	Number of Nodes	Activation Function
Input	12	ReLu
Fully Connected 1	12	ReLu
Dropout	N/A	N/A
Batch Normalization	N/A	N/A
Fully Connected 2	6	ReLu
Dropout	N/A	N/A
Batch Normalization	N/A	N/A
Fully Connected 3	4	ReLu
Dropout	N/A	N/A
Batch Normalization	N/A	N/A
Output	3	Softmax

**Table 4 sensors-22-07013-t004:** Participant demographics.

Participant	Sex (M/F)	Age Range (years)	Height (cm)	Weight (kg)	BMI (kg/m^2^)
1	M	70–79	171.5	70.0	23.9
2	F	70–79	160.0	56.8	22.1
3	F	70–79	170.0	61.3	21.2
4	M	30–39	180.3	84.1	25.9
5	F	50–59	157.0	85.0	34.5
6	M	20–29	167.0	69.5	24.5
7	F	60–69	156.0	61.4	25.2
8	F	60–69	162.0	54.4	20.7
9	M	60–69	181.0	80.5	24.5

**Table 5 sensors-22-07013-t005:** Combined mean accuracy and standard deviations of the tested models for classifying positions as left-side lying, supine, or right-side lying.

Model	c = 0%	c = 10%	c = 20%	c = 30%
ADA	62.27% ± 15.44%	66.68% ± 12.21%	71.86% ± 10.33%	74.37% ± 11.09%
GBC	62.27% ± 13.45%	79.09% ± 8.66%	84.98% ± 6.33%	90.13% ± 4.64%
LGB	61.68% ± 13.25%	93.94% ± 4.52%	96.86% ± 2.00%	97.99% ± 1.42%
LR	67.77% ± 16.01%	69.05% ± 15.05%	70.18% ± 14.34%	71.23% ± 13.81%
MLP	67.92% ± 14.95%	69.35% ± 17.90%	73.06% ± 15.29%	75.40% ± 10.68%
OG	64.11% ± 16.01%	78.25% ± 10.91%	85.69% ± 6.70%	88.58% ± 6.27%
SVM	71.25% ± 15.84%	72.21% ± 15.54%	73.09% ± 15.21%	74.01% ± 14.85%
XGB	61.75% ± 14.02%	93.40% ± 5.12%	96.83% ± 1.82%	98.12% ± 1.03%

**Table 6 sensors-22-07013-t006:** Combined F1 scores ± standard deviations of the tested models for classifying participant positions as left-side lying, supine, or right-side lying.

Model	c = 0%	c = 10%	c = 20%	c = 30%
ADA	0.654 ± 0.153	0.697 ± 0.1240	0.7436 ± 0.1100	0.766 ± 0.1190
GBC	0.650 ± 0.147	0.808 ± 0.0813	0.8637 ± 0.0578	0.910 ± 0.0369
LGB	0.639 ± 0.138	0.946 ± 0.0361	0.9713 ± 0.0152	0.981 ± 0.0116
LR	0.695 ± 0.168	0.710 ± 0.1540	0.7225 ± 0.1440	0.733 ± 0.1370
MLP	0.695 ± 0.153	0.708 ± 0.1880	0.7441 ± 0.1500	0.769 ± 0.1120
OG	0.662 ± 0.161	0.802 ± 0.0901	0.8728 ± 0.0494	0.898 ± 0.0470
SVM	0.713 ± 0.167	0.723 ± 0.1640	0.7317 ± 0.1590	0.741 ± 0.1540
XGB	0.637 ± 0.154	0.941 ± 0.0403	0.9707 ± 0.0144	0.982 ± 0.0030

**Table 7 sensors-22-07013-t007:** Results of the inter-rater reliability analysis using Fleiss’s kappa (κ).

Participant	Left-Side (κ)	Supine (κ)	Right-Side (κ)	Combined (κ)	*n*
1	0.998	0.997	0.988	0.991	1754
2	0.990	0.984	0.998	0.988	1684
3	-	0.970	0.975	0.945	1435
4	0.997	0.836	0.961	0.934	996
5	1.000	0.984	0.996	0.994	1419
6	0.999	0.989	1.000	0.995	1944
7	0.999	0.769	0.974	0.959	820
8	1.000	0.859	0.861	0.880	1295
9	1.000	1.000	1.000	1.000	737
Overall (κ)	0.960	0.969	0.948	0.935	12,084

## Data Availability

Not applicable.

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
