# Peer review of "Measuring Repositioning in Home Care for Pressure Injury Prevention and Management"

_sensors, 2022, doi:10.3390/s22187013_

Round 1

Reviewer 1 Report

TThe research activities are well structured, being of a great  importance in the field of rehabilitation and a future solution for mobility impaired subjects and caregivers. There is a growing need of solutions to help the caregivers work of caring and monitoring bedridden people with risk of pressure sores. The authors correctly describe the activities and the importance of applying a monitoring sensor system to help positioning and to reduce the risk of pressure sores.

However, the authors must consider:

-        To detail what is a pressure injury, the causes and why it is considered a devastating complication of immobilization. The authors must consider the use of more bibliographic references – lines 31 to 33

-        The authors refer “ It is not surprising that the costs of preventing a pressure injury are much lower than the costs to manage them”. Must use updated bibliographic references – lines 37 and 38

-        On line 39, authors refer “but not all pressure injuries are preventable”. This information must be explained for a better understanding of the study objective and the importance of sensors.

-        The use of different materials in mattresses and cushions, are strategies to relief pressure on bony areas and reduce the risk of pressure sores. The authors must consider this topic in the introduction to set the basis for future work /future perspectives (discussion) – lines 39 to 45. In future perspectives, the authors must consider the mattresses’ material, subject’s body mass and subject’s bony areas as study variables to improve the pressure sensors’ reliability.

Author Response

To detail what is a pressure injury, the causes and why it is considered a devastating complication of immobilization. The authors must consider the use of more bibliographic references – lines 31 to 33

We have added additional information and additional bibliographic references. "Pressure injuries also known as bedsores, are caused by damage to the skin and underlying soft tissue due to prolonged tissue deformation disrupting cell homeostasis leading to cell death and inflammatory edema [1]. Pressure injuries, most commonly occurring over a boney prominence can be a devastating secondary complication of immobility [2]. "

The authors refer “ It is not surprising that the costs of preventing a pressure injury are much lower than the costs to manage them”. Must use updated bibliographic references – lines 37 and 38

We have provided a more updated reference.

On line 39, authors refer “but not all pressure injuries are preventable”. This information must be explained for a better understanding of the study objective and the importance of sensors.

We have provided additional details. "Many, but not all pressure injuries are preventable [8]. For example, patients who are unable to maintain adequate nutrition or hydration, are unable to be repositioned due to hemodynamic instability, or are at the end of life may develop pressure injuries [8]."

The use of different materials in mattresses and cushions, are strategies to relief pressure on bony areas and reduce the risk of pressure sores. The authors must consider this topic in the introduction to set the basis for future work /future perspectives (discussion) – lines 39 to 45. In future perspectives, the authors must consider the mattresses’ material, subject’s body mass and subject’s bony areas as study variables to improve the pressure sensors’ reliability.

We have added additional details in the introduction "Pressure injury management requires a comprehensive interdisciplinary patient-centred approach that focuses on key behaviours including addressing nutrition, pressure offloading, maintaining and adequate moisture balance, ensuring the use of adequate support surfaces and tracking pressure injuries [7]."  In the discussion we have included other variables we would consider in future work ""We plan to undertake further training and testing our machine learning models using participants with varying bod mass, a wider range of sleeping positions with recommended offloading devices on individuals at risk of developing pressure injuries in future studies to improve the generalizability of our work and ensure our high accuracy is not the result of overfitting."

Reviewer 2 Report

Presented article with Title ” Measuring Repositioning in Home Care for Pressure Injury Prevention and Management ” is writing on 14 pages with 7 figures, 7 tables and 20references. This article is written clearly and comprehensibly. The article focuses on a practical example with a specific data. All data were verified oj 9 participant. The paper is worth publishing, but in my opinion, the manuscript content dont needs some improvements.

Suggestions:

-    Chapter 1. introduction is too short.

·      -   Chapter 2. is described too clearly and distinctly.

·     -    Please provide a more detailed description of the technical equipments.

All the specific comments can be followed in reviewed copy of the manuscript.     

I recomend this paper publish in journal after minor revision.

Author Response

Chapter 1. introduction is too short

We have added additional detail in the introduction. Please refer to the TRACK changes document.

Chapter 2. is described too clearly and distinctly.

We are unclear as to what recommendations the reviewer is suggesting.

Please provide a more detailed description of the technical equipments.

We have added additional detail in the equipment section. We have included that we used MATLAB to process the data. 

Round 2

Reviewer 1 Report

The authors clarified the information and detailed the need of a new technology for pressure ulcer prevention. The bibliographic references are updated. The research activities are well structured, being of a great importance in the field of rehabilitation and a future solution for mobility impaired subjects and caregivers. There is a growing need of solutions to help the caregivers work of caring and monitoring bedridden people with risk of pressure sores.